# The Geographic Context of Racial Disparities in Aggressive Endometrial Cancer Subtypes: Integrating Social and Environmental Aspects to Discern Biological Outcomes

**DOI:** 10.3390/ijerph19148613

**Published:** 2022-07-15

**Authors:** Anna Kimberly Miller, Jennifer Catherine Gordon, Jacqueline W. Curtis, Jayakrishnan Ajayakumar, Fredrick R. Schumacher, Stefanie Avril

**Affiliations:** 1Department of Genetic and Genome Sciences, Case Western Reserve University, Cleveland, OH 44106, USA; akm126@case.edu; 2Department of Obstetrics and Gynecology, Division of Gynecologic Oncology, University Hospitals Cleveland and Case Western Reserve University, Cleveland, OH 44106, USA; jennifer.gordon@uhhospitals.org; 3Case Comprehensive Cancer Center, Cleveland, OH 44106, USA; frs2@case.edu; 4Department of Population and Quantitative Health Sciences, Case Western Reserve University, Cleveland, OH 44106, USA; jxa421@case.edu; 5Department of Pathology, University Hospitals Cleveland and Case Western Reserve University, Cleveland, OH 44106, USA

**Keywords:** endometrial cancer subtypes, racial disparities, geospatial, environmental mechanism, social vulnerability

## Abstract

The number of Endometrial Carcinoma (EC) diagnoses is projected to increase substantially in coming decades. Although most ECs have a favorable prognosis, the aggressive, non-endometrioid subtypes are disproportionately concentrated in Black women and spread rapidly, making treatment difficult and resulting in poor outcomes. Therefore, this study offers an exploratory spatial epidemiological investigation of EC patients within a U.S.-based health system’s institutional cancer registry (*n* = 1748) to search for and study geographic patterns. Clinical, demographic, and geographic characteristics were compared by histotype using chi-square tests for categorical and t-tests for continuous variables. Multivariable logistic regression evaluated the impact of risks on these histotypes. Cox proportional hazard models measured risks in overall and cancer-specific death. Cluster detection indicated that patients with the EC non-endometrioid histotypes exhibit geographic clustering in their home address, such that congregate buildings can be identified for targeted outreach. Furthermore, living in a high social vulnerability area was independently associated with non-endometrioid histotypes, as continuous and categorical variables. This study provides a methodological framework for early, geographically targeted intervention; social vulnerability associations require further investigation. We have begun to fill the knowledge gap of geography in gynecologic cancers, and geographic clustering of aggressive tumors may enable targeted intervention to improve prognoses.

## 1. Introduction

Endometrial carcinoma (EC) is the most common cancer of the female genital tract [1]. It is categorized into five frequent histological subtypes (histotypes), which describe the histological characteristics and biological behavior of the tumor: endometrioid, serous, mixed, and clear-cell carcinoma, and carcinosarcoma [2]. While endometrioid is the most common histology, representing about 75% of all EC, histotype-specific cancer incidence differs by population, where Black women in the US have lower survival and are disproportionately diagnosed with the aggressive non-endometrioid histotypes [2,3]. Prior risk stratification systems show that Black women in the U.S. have poorer outcomes, as they are diagnosed at a later stage [4] with higher grade [5] and more aggressive non-endometrioid histology [6]. Although Black women often present with more advanced stages of disease, higher-grade tumors, and aggressive histotypes, their survival is significantly lower for all histotypes even after stratifying by age, stage, and grade [7,8,9].

To improve EC incidence, treatment, and severity outcomes, prior studies on racial disparities have only analyzed biological factors, such as age or Body Mass Index (BMI), in isolation. For example, age is associated with cancer risk, where older individuals are overall at an increased risk for cancer, including EC [10]. Obesity has inflammatory components that are related to many major chronic diseases, including endometrial cancer [8]. BMI is a strong risk factor for the most prevalent endometrioid histotype, where higher BMI tends to be associated with lower-grade and earlier-stage tumors but is a weaker and more contradicted risk factor for the less prevalent but more aggressive non-endometrioid histotype [11]. While the focus on biological risks is essential, it does not fully explain the racial disparities that are observed. Another potential driver of disparities is access to healthcare. However, there is some evidence that demonstrates that this too is not enough to explain the racial differences that exist [8,12,13]. What else may be driving these outcomes? Furthermore, of these other potential risks, what is detectable such that it can be modified or included in risk stratification for patients so that clinicians can provide improved personalized care [14,15]? In the United States, it is known that there exists a racially patterned landscape, such that Black people are disproportionately exposed to a range of carcinogenic social and environmental risks [13,16,17]. How might this landscape be associated with tumor biology in endometrial cancer? This study presents an exploratory spatial epidemiological investigation of this question, as currently, the interactions of biological, sociodemographic, and environmental mechanisms driving the outcome disparities remain poorly understood [15]. 

Spatial epidemiology offers geographically contextualized approaches to measure dimensions of small areas, such as neighborhoods, and of individuals within these areas. Such analyses usually rely on mapping patient residential addresses as points on a map in a Geographic Information System (GIS) and then adding other relevant layers, ranging from community level social determinants of health to individual locations of risks (e.g., violent crime, hazardous facilities). This geographic context surrounding where a patient with endometrial cancer lives can then be quantified and integrated in modeling with their individual-level characteristics [18]. GIS has been widely utilized in cancer epidemiology, but less so specifically in the study of endometrial cancer [19,20]. Furthermore, even fewer studies have used GIS to investigate spatial patterns of potential etiological variables in tumor biology. Instead, most have focused on other aspects of the Cancer Control Continuum, such as prevention, early detection, diagnosis, etc. [21]. Recently, however, there has been growing interest in the neighborhood-level associations linked to racial disparities in breast cancer biology, particularly the role of neighborhood deprivation and molecular mechanisms [22]. Building from this framework, this study explored associations between endometrial tumor biology and two widely utilized, and widely accessible, geographic datasets that can be proxies for different aspects of neighborhood deprivation: the Environmental Protection Agency’s (EPA’s) Toxic Release Inventory (TRI) [23] and the Centers for Disease Control and Prevention (CDC) Social Vulnerability Index (SVI) [24,25]. The TRI is a dataset of facilities that handle toxic chemicals monitored by the EPA. Each record lists the amounts of all reportable chemicals at a geographic location, including the x,y coordinate and all associated variables (For a complete list of variables reported in the TRI, see: https://www.epa.gov/system/files/documents/2021-10/tri-basic-data-file-documentation-ry2020_100721.pdf, accessed on 19 January 2022), and can be mapped using GIS. The TRI has been widely used in cancer literature to investigate the role of environmental hazards on disease [26,27,28]. While the TRI can be used to investigate chemical exposure and disease outcome, drawing from environmental justice and environmental racism research, it can also be used as a marker of racially patterned aspects of neighborhood deprivation [29]. In this study, we used this dataset to explore if and perhaps how it may be associated with tumor biology. The second exploratory dataset we used was the SVI. It is a composite area-based measure of 15 census variables at the county and census tract levels covering four themes: socioeconomic status, household composition and disability, minority status and language, and housing type and transportation. The cancer literature has used SVI to assess the resiliency and vulnerability of different communities to external stress [30,31]. In this study, it was used as a marker of the geographic context of socioeconomic conditions in the patients’ residential area. Overall, associations between tumor type and geographic context, such as neighborhood deprivation, have not been studied in EC. We suggest that markers of neighborhood deprivation risks can be associated with histotype and the underlying biology. We sought to examine the association of endometrioid versus non-endometrioid histology and survival with an individual’s social and physical environment.

## 2. Materials and Methods

### 2.1. Study Population

We collected data from 4039 patients diagnosed with endometrial cancer, including endometrioid and non-endometrioid subtypes, between 1998 and 2021 at University Hospitals Cleveland (UH) Seidman Cancer Center from the institutional cancer registry, with follow-up for 22 years. All patients were treated with primary surgery including hysterectomy and bilateral salpingo-oophorectomy. Additionally, Patients considered to be at high-risk for recurrence, including those with non-endometrioid subtypes or advanced stage (FIGO stage II or III), were offered adjuvant chemotherapy or pelvic radiation. These individuals were matched with the Ohio Cancer Incidence Surveillance System (OCISS) from 1992 to 2018 in order to obtain outcome data for overall and cancer-specific survival. Records were excluded if they did not have a complete street address and, of these, if the residential address listed was outside of Ohio. The fifteen counties comprising the Case Comprehensive Cancer Center (CCCC) catchment area were used to define the study area. All residential addresses that were located within these counties were geocoded and within this dataset, addresses that did not have high positional accuracy were removed, resulting in a dataset of 3582 women. Only 1748 women could be linked by diagnosis date between the two datasets, and 836 were excluded due to missing data (Appendix A). While we were unable to use the full original dataset, the proportions of race (Black or White women) and disease stage (stage I or higher stage) are similar between both datasets.

### 2.2. Demographic Characteristics

Relevant variables available from our institutional cancer registry include histology, height, weight, FIGO stage, race, and residential addresses (subsequently geocoded to X-, Y-coordinates). Patients were categorized as having endometrioid versus non-endometrioid histotypes. The non-endometrioid category comprises several histotypes, including serous, clear cell, or mixed carcinoma, and carcinosarcoma.

The OCISS database contains statewide cancer surveillance data for incidence and survival of cancer by the National Program of Cancer Registries. All cancer cases diagnosed among Ohio residents, except for basal cell and squamous cell carcinoma of the skin, are reported to OCISS. The OCISS database follows the data standards and data dictionary of the North American Association of Central Cancer Registries and includes clinical characteristics at diagnosis (stage, grade, and cancer site), demographics (age, sex, and race), date of diagnosis, primary treatment information, vital status follow-up, and cause of death. 

### 2.3. Geospatial Analyses

Patient residential addresses with complete model data were available for 912 unique patients, with high positional accuracy. Each individual was geocoded to their point level locations. Spatial epidemiological cluster detection techniques, Local Moran’s I (LMI) [32], Gi* [33], and GeoMEDD [34] were used to investigate geographic patterns of non-endometrioid subtypes within the CCCC area in ArcMap (v = 10.7.1) [35]. Local Moran’s I and Gi* have been used widely in detection of cancer clusters [19,28,36,37], while GeoMEDD is a new approach developed in COVID-19 spatial syndromic surveillance [34]. Local Moran’s I and Gi* operate on aggregate geographic units, such as census tracts, in this case. GeoMEDD output is based on the boundary of point level cases based on user-defined space–time relationships, which identify granular concentrations, such as at a street segment or building level. While GeoMEDD reports only the presence of geographical areas that meet user-specified spatial and temporal relationships (but not significance thresholds), LMI and Gi* compare the observed spatial pattern of disease with the null hypothesis of complete spatial randomness (CSR) as generated by 999 Monte Carlo simulations [28,38]. Local Moran’s I and Gi* were calculated for the rate of non-endometrioid subtypes by census tract, with neighbors defined as having shared boundaries and vertices (1st order queen contiguity matrix). The output from these techniques is multiple clusters of varying sizes and significance thresholds of *p* ≤ 0.01. For example, some clusters may be *p* = 0.01, while others may be *p* = 0.0025, etc. Therefore, results are reported for all clusters as *p* ≤ 0.01. It should also be noted that these significance values are pseudo *p*-values, as spatial cluster detection inherently relies on multiple testing procedures [39]. 

Neighborhood deprivation proximate to each address was integrated within a Geographic Information System (GIS). The Centers for Disease Control and Prevention (CDC) Social Vulnerability Index (SVI) approximated social conditions using 15 census variables at the census tract level (Figure 1) [24]. Each census tract was ranked by percentiles ranging from 0 to 1 with greater values indicating greater social vulnerability [24]. For each patient, an 800-m buffer was drawn circling their address at diagnosis and the average SVI score for the year 2018 was individually coded. The Environmental Protection Agency’s (EPA) Toxic Release Inventory (TRI) served as another marker of risk in the home environment of each patient [23]. Geographic X, Y coordinates pinpointed each facility’s location. Using the 800-m buffer, the number of TRI facilities as well as the mean and median chemical release values reported by these facilities for the years 2000, 2010, and 2020 were calculated for each patient (Figure 1). This buffer size was selected for this exploratory analysis based on its widespread use as a walkable distance in numerous studies on utilization of the neighborhood environment [40,41]. 

### 2.4. Statistical Analysis

Clinical, demographic, and environmental characteristics were compared by histotype (endometrioid versus non-endometrioid) using chi-square tests for categorical variables (i.e., FIGO stage and race) and t-tests for continuous variables (i.e., age, BMI, SVI, the total number of TRI releases (TRI density), and the number of TRI facilities (TRI count)). Normality was measured using both visual density and Q-Q plots as well the Shapiro-Wilk statistical approach. We categorized the variables SVI, TRI density, and TRI count for our modeling of outcomes. SVI quartiles were utilized, where the highest quartile captured the greatest social vulnerability. Both TRI variables were categorized as having no facilities or releases (0) or any facilities or releases (>0). Multivariable logistic regression models were used to evaluate the impact of health, social, and physical environmental risks of histotypes. All multivariable models were adjusted for age, BMI, FIGO stage, and race.

A survival analysis employing Cox proportional hazard models and a log-rank test compared risk of overall death (219 deaths, 693 survival) and cancer-specific survival (72 deaths, 840 survival). Both univariate and multivariable analyses were employed for survival outcomes, where multivariable models were adjusted for age at diagnosis (continuous), BMI (obese vs. non-obese), FIGO stage (stage I vs. higher stage), and race (Black vs. White). All statistical analyses were performed using R and two-sided statistical tests were employed, where statistical significance was defined applying a threshold of *p* = 0.05. Given this analysis was exploratory, multiple comparisons were not considered.

## 3. Results

Among the 912 women with endometrial cancer, 649 (71.2%) had the endometrioid subtype and 263 (28.8%) had non-endometrioid subtypes (Table 1). Chi-square tests and *t*-tests reported significant differences by histotype for age, SVI, TRI density, BMI, FIGO stage, and race. On average, individuals with the non-endometrioid subtype were older, had a higher SVI, a lower density of TRI releases, a lower BMI, and a more advanced FIGO stage (stage II–IV), and were predominantly Black women (Table 1). 

The three geospatial techniques identified similar statistically detectable geographic clusters of non-endometrioid histotype rates located in two high social vulnerability areas of Cuyahoga County as well as in more rural areas of northeast Ohio (*p* < 0.05). GeoMEDD identified clusters ranging in size from areas of a neighborhood to individual congregate living facilities. Furthermore, there were five individual addresses with at least three patients who had non-endometrioid subtype tumors. These are not duplicate records, rather they are indicative of congregate living facilities such as nursing homes and apartments. In some census tracts with low populations, it can be these individual point locations that drive the rate for the tract. In these cases, it is not neighborhood-level conditions that may be associated with the more aggressive tumor subtype, but rather women at older ages concentrated by buildings. 

The results from a multivariable logistic regression model are presented in Table 2 and were adjusted for covariates (Table 1). The models measured the association of endometrioid vs. non-endometrioid histotype with either the social variable, SVI, or the environmental variables, TRI count and TRI density, adjusted for age, BMI, FIGO stage, and race. SVI was associated with histotype both as a continuous variable (OR = 2.14; 95% CI = 1.26, 3.63; *p* = 4.70 × 10^−3^; data not shown) and as a categorical variable (OR = 1.77; 95% CI = 1.16, 2.72; *p* = 0.008) comparing high SVI to the reference, low SVI (Table 2 and Appendix A). Under the categorical variable model, women with the greatest social vulnerability index have a 77% increased risk of non-endometrioid EC compared to women residing in areas with minimal social vulnerability. Neither of the TRI variables were significantly associated with endometrioid or non-endometrioid histotypes in these multivariable models (Table 2 and Appendix A).

A survival analysis evaluated the time to overall death for the social and environmental variables SVI, TRI count, and TRI density as continuous and categorical variables. In models similar to the multivariable logistic regression model, each of these variables, SVI, TRI count, and TRI density, were individually measured with adjustment by covariates age, BMI, FIGO stage, and race. SVI was associated with survival both as a continuous variable (HR = 1.86; 95% CI = 1.17, 2.94; *p* = 8.33 × 10^−3^) and as a categorical variable comparing high SVI to low SVI (HR = 1.67; 95% CI = 1.14, 2.41; *p* = 7.95 × 10^−3^) (Figure 2 and Appendix A, Table 3 and Appendix A). Neither of the TRI variables were significantly associated with survival in these models (Table 3 and Appendix A).

Additionally, multivariable survival analyses predicted increased likelihood of overall deaths with SVI. The association of total survival and SVI was significant in models with SVI as a continuous or a categorical variable (Appendix A). Those with the high SVI category were significantly associated with death compared to those with the low SVI category. The endometrial cancer-specific deaths model did not replicate the results, which may be due to a smaller sample size of cancer-specific deaths (Appendix A).

## 4. Discussion

We used a modeling approach to evaluate the association between endometrial cancer histotype and several community-scale factors for EC cases diagnosed in the CCCC area. A model was built to assess the relationship of endometrioid versus aggressive/non-endometrioid histotype and a novel variable, SVI, adjusting for age, BMI, FIGO stage, and race. SVI was associated with the aggressive non-endometrioid histotype as both a continuous and a categorical variable. For example, a woman residing in an area with the highest SVI quartile has a 77% increased risk for developing non-endometrioid cancer compared to the women living in the area with the lowest SVI quartile. In a race-stratified analysis, SVI is significantly associated with histotype as both a continuous and categorical variable in White women (data not shown). No significant association was detected in Black women, as the current analysis is underpowered, but the SVI odds ratios were in the same direction as the analysis with only White women. Given the impact of histotype on progression, social vulnerability may be indicative of disease severity and should be considered in the treatment of all women. While other studies have not detected an association between tumor type and SVI, SVI has been associated with poorer post-surgery outcomes [42] and less access to treatment [43,44] in cancer patients.

We also geographically clustered endometrial histotype in the CCCC area. Both endometrioid and non-endometrioid histotype clusters were detected and indicated that the CCCC area has a racially patterned external environment, as is the case throughout much of the U.S. [13]. This is the first time (to our knowledge) that endometrial histotype has been geographically clustered. The mapped output will be used internally to target clinical intervention and to direct more locally targeted analyses. The SVI data can be used to create risk stratifications based on where individuals live for more holistic treatments. Those residing in high-risk areas could receive extra screening that may lead to earlier diagnosis. Targeted screening could improve health outcomes of individuals with non-endometrioid EC, as there are fewer early diagnosable symptoms. 

Follow-up studies using a greater population of endometrial-cancer-specific deaths are needed to confirm the survival analysis results. Overall, social vulnerability should be considered in treatment, as it may also be indicative of overall survival. We expect that improved screening and targeted treatment derived from SVI risk stratifications could decrease mortality rates, particularly from the high-risk non-endometrioid histotypes [45,46,47,48]. 

While there are significant associations between SVI and both non-endometrioid EC and survival, the biological mechanism is still unknown. High social vulnerability may be associated with biological stress responses, as the cancer literature has previously used SVI to assess the community’s resiliency and vulnerability from external stressors [30,31,49,50,51]. Future studies could better define why Black women and women residing in areas of high SVI are more frequently diagnosed with non-endometrioid EC. There could be a stress-related biological mechanism, such as changes in DNA methylation or host immune response, that is driving the increase in non-endometrioid EC diagnosis. Additionally, a particular component encompassing SVI could drive the association in different communities. A geographically weighted regression of histotype and SVI could detect distinct clusters where SVI is more or less predictive [52]. As we were unable to use the full dataset due to a lack of linkage between institutional and Ohio cancer databases based on diagnosis dates, we may be able to detect more distinct clusters with a fuller dataset. Further studies could also detect associations between each of the 15 components encompassing the SVI measurement to understand if different clusters of histotype are associated with specific vulnerabilities.

No significant association was detected between either TRI variable, continuous or categorical, and histotype or death. This may be due to the limited sample size of individuals with TRI facilities or releases greater than zero. Some women in our sample resided in regions with TRI facilities and releases greater than zero but not enough to detect a statistical difference (Appendix A). It may also be that environmental exposures are not from TRI facilities, as they are regulated, but from non-reporting facilities, which may pose more of a risk to health [16,53]. Future studies in areas with a variety of TRI exposures are needed to measure if there is a relationship between TRI facilities or releases and histotype or death.

## 5. Limitations

This study offers novel insights into the geographic context of tumor biology in endometrial cancer and spatial methods to search for granular places that can be targeted for education and intervention; however, it has a number of limitations that should be considered. Some of these will require further investigation as this line of inquiry develops. First, the study does not account for the temporal aspect of the exposure–outcome relationship. We do not know how long the patients lived in their home address nor if the geographic context of their surroundings in this placed changed over time, specifically in the chosen proxies for neighborhood deprivation. We also assume stationarity in the patient home addresses, which may not be the case. There is evidence to demonstrate that low-income patients in particular exhibit elevated residential mobility, often forced mobility as in the case of evictions. Furthermore, we lack knowledge of daily mobility of patients and their surroundings, such as areas where they spend their time away from the home address that may be more relevant sources of exposure to risks such as TRI. Despite its importance for measuring exposure, such “activity space” geography is not a part of the Electronic Medical Record (EMR). These limitations are a part of a larger concern, the Uncertain Geographic Context Problem (UGCoP) [54], which our team is addressing [55,56,57]. Furthermore, we lacked a replication site. Replicate studies in other cities with a history of a racially patterned environment may be needed to further support our claims [58,59]. In particular, recent studies have questioned the role of neighborhood, especially how the characteristics of deprivation, stressors, and social vulnerability influence biological processes that lead to adverse health outcomes, such as DNA methylation [60]. This study was limited to women who self-identified as Black or White. Our results cannot generalize to women of other races and ethnic groups without further analyses.

## 6. Conclusions

The United States displays a racially patterned external environment that likely impacts health and survival of women with EC. Individuals in our study with the more aggressive non-endometrioid subtype were predominantly Black women and resided in areas with higher social vulnerability. These associations represent an opportunity for improved targeted screening and risk stratification based on a woman’s health characteristics and geographic location. Clinical risk stratification using SVI and other risk variables can detect who benefits most from extra screening and care. Clinicians may be able to improve prediction and detection of aggressive EC and decrease poor outcomes if social and physical factors are included as risk factors.

## Figures and Tables

**Figure 1 ijerph-19-08613-f001:**
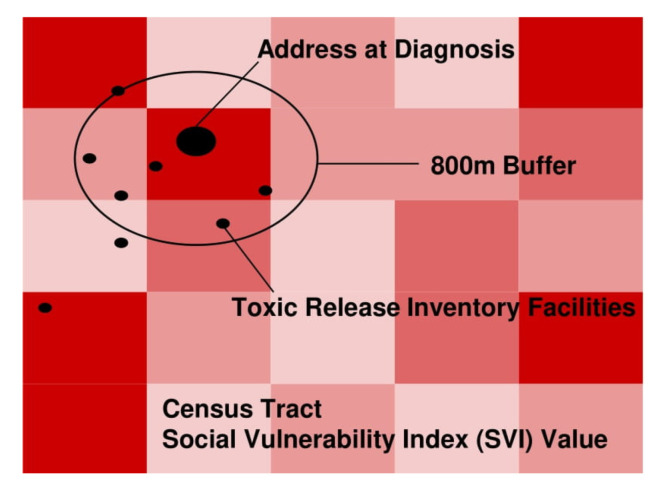
For each patient, an 800-m buffer was drawn circling their address at diagnosis and the average SVI score for the year 2018 was individually coded. The Environmental Protection Agency’s (EPA’s) Toxic Release Inventory (TRI) serve as another marker of risk in the home environment of each patient [23]. Geographic X, Y coordinates pinpoint each facility’s location. Using the 800-m buffer, the number of TRI facilities as well as the mean and median chemical release values reported by these facilities for the years 2000, 2010, and 2020 were calculated for each patient.

**Figure 2 ijerph-19-08613-f002:**
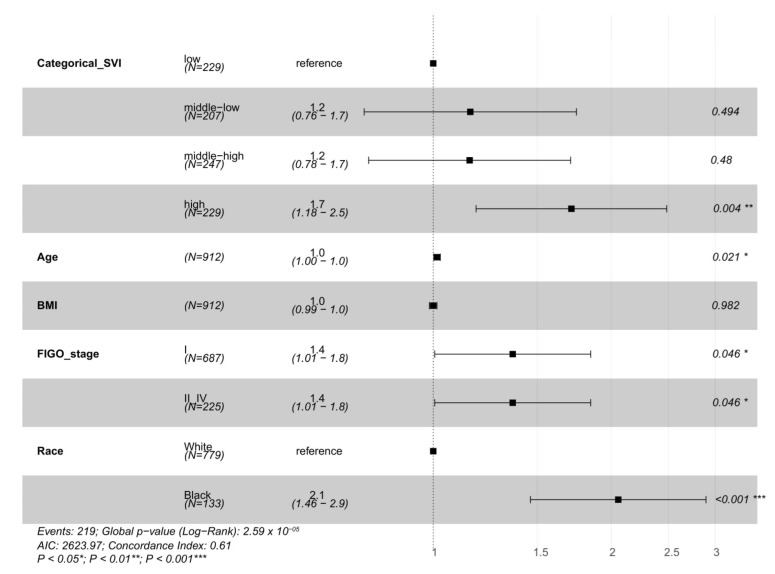
Overall deaths survival analysis hazard ratios. In the overall deaths survival analysis, SVI high is significant (*p* = 0.008) when adjusting for age, BMI, FIGO stage, and race. Advanced age, higher BMI, and Black women have odds ratios greater than or equal to 1 and therefore are at an increased risk of death.

**Table 1 ijerph-19-08613-t001:** Demographics of study population by histotype, mean (SD).

	Endometrioid (N = 649)	Non-Endometrioid (N = 263)	*p*-Value ^1^
**Age**	62.63 (10.40)	65.37 (10.80)	5.03 × 10^−4^ *
**SVI**	0.42 (0.28)	0.47 (0.30)	0.01 *
**TRI Count**	0.35 (1.43)	0.51 (3.05)	0.41
**TRI Density**	881.15 (6741.13)	220.138 (1493.28)	0.02 *
**BMI (mean (range))**	35.51 (13.13–87.58)	33.39 (16.95–79.53)	2.82 × 10^−3^ *
**FIGO stage, stage 1 count (%)**	527 (81.20%)	160 (60.84%)	1.79 × 10^−10^ *
**Race**			4.28 × 10^−10^ *
Black	64 (10.12%)	69 (25.57%)	
White	585 (89.88%)	194 (74.43%)	

^1^ A *t*-test was performed for age, SVI, TRI count, TRI density and BMI. A chi-square test was performed for FIGO stage and race. * *p*-value < 0.05.

**Table 2 ijerph-19-08613-t002:** Multivariable Models that Associate Histotype with Categorical Variables.

	Full Model SVI			Full Model TRI Count			Full Model TRI Density	
	OR (95% CI)	*p*-Value		OR (95% CI)	*p*-Value		OR (95% CI)	*p*-Value
**SVI Middle-Low**	1.25 (0.80, 1.95)	0.34	**TRI Count High**	0.85 (0.48, 1.47)	0.58	**TRI Density High**	0.99 (0.51, 1.84)	0.98
**SVI Middle-High**	1.00 (0.64, 1.55)	0.99						
**SVI High**	1.77 (1.16, 2.72)	0.008 *						

Adjusted for age, BMI, FIGO stage, and race. * *p*-value < 0.05.

**Table 3 ijerph-19-08613-t003:** Overall Deaths Multivariable Analysis with Categorical Variables.

	Full Model SVI			Full Model TRI Count ^1^			Full Model TRI Density ^2^	
	Hazard Ratio (95% CI)	*p*-Value		Hazard Ratio (95% CI)	*p*-Value		Hazard Ratio (95% CI)	*p*-Value
**SVI Middle-Low ^3^**	1.15 (0.76, 1.73)	0.51	**TRI Count High**	0.84 (0.51, 1.35)	0.46	**TRI Density High**	0.97 (0.57, 1.68)	0.93
**SVI Middle-High ^4^**	1.08 (0.73, 1.62)	0.69						
**SVI High ^5^**	1.67 (1.14, 2.41)	7.95 × 10^−3^ *						

Adjusted for age, BMI, FIGO stage, and race. * *p*-value < 0.05. ^1^ Patient Survival low = 630, Patient Death low = 201, Patient Survival high = 63, Patient Death high = 18. ^2^ Patient Survival low = 649, Patient Death low = 205, Patient Survival high = 44, Patient Death high = 14. ^3^ Patient Survival = 162, Patient Death = 45, ^4^ Patient Survival = 192, Patient Death = 55, ^5^ Patient Survival = 156, Patient Death = 73.

## Data Availability

Not applicable.

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
