# Peer review of "The Geographic Context of Racial Disparities in Aggressive Endometrial Cancer Subtypes: Integrating Social and Environmental Aspects to Discern Biological Outcomes"

_ijerph, 2022, doi:10.3390/ijerph19148613_

Round 1

Reviewer 1 Report

The authors have conducted a geographic context analysis of women treated for endometial cancer and evaluated the impact of spatial epidemiological as well as known biological risk factors. This is an interesting subject of possible clinical as well as societal importance.

The introduction explain well the traditional risk factors (age, endometrioid vs non-endometrioid, FIGO stage, BMI) as well as known racial disparities in a US context.

The method of geospatial analysis is explained sufficiently.

What I do miss is a note in the statistical section that log-rank test have been used (assuming this is used comparing survival curves Fig S2)

In result section please state length of follow-up as well as number of deaths during follow up (events) to ensure that the multivariate analyses encompass «the rule of 10» : at least 10 events per variable included in the multivariate model. In table 3 number of patients and events should also be stated. In this model there is three categories for SVI and not stated how many factors adjusted for (age, BMI, FIGO stage, race, histologic type?) This need to be described in the table legend.   Each table/figure should be able for the reader to understand without reading the full paper.

FigS2 please state number of patients and events (death) for each of the four categories.

Please do not introduc results in the discussion that has not been presented in the result section (S3 line 291).

One more limitation of this study is that type of treatment has not been mentioned. (surgery, adjuvant chemo/radiation).

I do understand the limitation of geographical adress at time of diagnosis/treatment. 

Since the TRI analyses has been applied regarding living adress, could perhaps work environment be another TRI factor? These data (work adress) would probably not be available for this study but could perhaps be one factor why living adress TRI not being of significance?

Reviewer 2 Report

This manuscript is entitled: The geographic context of racial disparities in aggressive endometrial cancer subtypes: Integrating social and environmental aspects to discern biological outcomes, integrated the social vulnerability index (SVI) associated with non-endometrioid histotypes as both a continuous and categorical variable, independently of known risk factors. From an operational perspective, this study provides a methodological framework for any health system to adopt 40 for early, geographically targeted education and intervention. The authors’ search for non-endometrioid subtypes were predominantly Black women who resided in areas with higher social vulnerability. Overall, the proposed research is of interest with good potential. The authors carried out detailed studies to prove the concept. There are some questions and suggestions that may need to solve as below:

1.     Authors should provide an abbreviations list to make the entire manuscript easier to read.

2.     The SVI and TRI calculations data using an 800 M Buffer and census tract-level data in Figure 1 seems to need a quotation of the calculation formula to demonstrate the design of the geographic map.

3.     The authors should compile a table on whether other studies have reported on this statistical approach to analyze the modality of the number of women with endometrial cancer (EC) and social vulnerability.

4.     The conditions of the subjects should be listed in more detail—for example, age, health, skin color, BMI, etc.

5.     Whether the data in the table are statistically relevant, such as p-value, standard deviation, etc., such data, if available, should be indicated on the table to provide more statistical confidence.

Reviewer 3 Report

The manuscript entitled “The geographic context of racial disparities in aggressive endometrial cancer subtypes: Integrating social and environmental aspects to discern biological outcomes” is well written and have potential interest to IJERPH readers. A few minor improvements should be made. Authors need to reduce the abstract, and in line 236 (page 6), Table 1 should be replaced by Table 2. The Authors should also indicate in Statistical Analysis if they tested data normality and with what test.

Round 2

Reviewer 2 Report

The authors have resolved most of the concerns proposed by the reviewer, and the manuscript has been improved significantly. Therefore, we do not have further revision requirements for this updated manuscript.